# Evaluation of the Implementation of the Dutch Breast Cancer Surveillance Decision Aid including Personalized Risk Estimates in the SHOUT-BC Study: A Mixed Methods Approach

**DOI:** 10.3390/cancers16071390

**Published:** 2024-03-31

**Authors:** Jet W. Ankersmid, Ellen G. Engelhardt, Fleur K. Lansink Rotgerink, Regina The, Luc J. A. Strobbe, Constance H. C. Drossaert, Sabine Siesling, Cornelia F. van Uden-Kraan

**Affiliations:** 1Department of Health Technology and Services Research, University of Twente, 7522 NB Enschede, The Netherlands; s.siesling@utwente.nl; 2Santeon, 3584 AA Utrecht, The Netherlands; e.engelhardt@santeon.nl (E.G.E.);; 3ZorgKeuzeLab, 2611 BN Delft, The Netherlands; 4Department of Surgery, Canisius Wilhelmina Hospital, 6532 SZ Nijmegen, The Netherlands; 5Department of Psychology, Health & Technology, University of Twente, 7522 NB Enschede, The Netherlands; c.h.c.drossaert@utwente.nl; 6Department of Research and Development, Netherlands Comprehensive Cancer Organisation, 3501 DB Utrecht, The Netherlands

**Keywords:** breast cancer, post-treatment surveillance, follow-up, shared decision-making, PtDA, OUTcome information, risk communication

## Abstract

**Simple Summary:**

This study examined the use of the Breast Cancer Surveillance Decision Aid (BCS-PtDA) in eight Dutch hospitals. This PtDA supports information provision (including information about personalized recurrence risks) and decision-making about post-treatment surveillance. Health care professionals (HCPs) acknowledged that the tool helped make patients aware of their options but felt it increased their workload without clear benefits. While the tool was effective in presenting choices to patients, deliberation about the options was scarce. The main challenges were related to the extra time required and HCPs’ perception of the tool’s value. Risk communication was deemed generally adequate. The study suggests that, while the PtDA offers benefits, better integration and communication strategies are needed to enhance shared decision-making processes. In conclusion, the implementation of the BCS-PtDA led to choices being offered to patients. However, there is room for improvement in information provision and the application of shared decision-making.

**Abstract:**

Background: To improve Shared decision-making (SDM) regarding personalized post-treatment surveillance, the Breast Cancer Surveillance Decision Aid (BCS-PtDA), integrating personalized risk information, was developed and implemented in eight hospitals. The aim of this mixed-methods study was to (1) assess the implementation and participation rates, (2) identify facilitators and barriers for use by health care professionals (HCPs), (3) quantify the observed level of SDM, and (4) evaluate risk communication and SDM application in consultations. Methods: Implementation and participation rates and patients’ BCS-PtDA use were calculated using hospital registry data and BCS-PtDA log data. HCPs’ perspective on facilitators and barriers were collected using the MIDI framework. Observed SDM levels in consultation transcripts were quantified using the OPTION-5 scale. Thematic analysis was performed to assess consultation content. Results: The average PtDA implementation and participation rates were, respectively, 26% and 61%. HCPs reported that the PtDA supported choice awareness. Reported barriers for implementation were mainly increased workload and a lack of perceived benefits. The consultation analysis (*n* = 64) showed patients were offered a choice, but deliberation was lacking. Risk communication was generally adequate. Discussion: When the BCS-PtDA was used, patients were clearly given a choice regarding their post-treatment surveillance, but information provision and SDM application can be improved.

## 1. Introduction

Follow-up after curative treatment for early-stage breast cancer focuses on (1) managing lasting physical and/or psychosocial effects or complications resulting from treatment (aftercare) and (2) identifying locoregional recurrences (LRRs) or secondary primary tumors (SPs) (post-treatment surveillance). Despite varying risks for LRRs and SPs [1,2], post-treatment surveillance is one-size-fits-all: Dutch clinical guidelines recommend an annual physical breast exam and imaging (i.e., mammogram or MRI) for a period of at least five years [3]. Tailoring post-treatment surveillance can decrease patient and health care burden.

Women with a low risk for LRRs and/or SPs could safely choose to reduce the intensity of post-treatment surveillance, because for them, less intensive surveillance is as effective as more intensive surveillance in terms of prognosis [4,5]. However, the impact of surveillance on quality of life can differ. For some, recurring surveillance appointments and waiting for test results can induce cancer worry, whilst, for others, less intensive surveillance can cause increased cancer worry. This makes decisions about post-treatment surveillance preference-sensitive [6]. Shared decision-making (SDM) is a process in which patients and health care professionals (HCPs) work together to make decisions about the patient’s care. SDM consists of four steps: (1) creating choice awareness; (2) presenting information about the benefits and harms of medically viable options (including the option to deescalate or forego treatment/diagnostics); (3) deliberation about the patient’s values, preferences, and circumstances; and (4) decision-making [7]. Even though SDM is widely advocated, its application in clinical practice remains suboptimal [8].

Tools such as Patient Decision Aids (PtDAs) and predictive and prognostic models can help to support SDM. PtDAs are evidence-based tools to help patients understand options, potential benefits and harms, including associated risks and uncertainties. PtDAs provide patients with unbiased information, facilitate discussion between patients and HCPs, and help patients clarify values and preferences. There is evidence that SDM with a PtDA can improve patient satisfaction, adherence to treatment, and support the choice of more conservative treatment options [9]. The Breast Cancer Surveillance Decision Aid (BCS-PtDA) was developed to support decision-making for four separate decisions regarding the organization of post-treatment surveillance for breast cancer survivors: (1) surveillance frequency, (2) duration, (3) examinations, and (4) how results of examinations will be discussed [10]. The BCS-PtDA incorporates personalized risk information from the INFLUENCE 2.0 nomogram [1,2], a validated prognostic model that provides personalized estimates of the 5-year risk for developing LRRs, SPs, and distant metastases during post-treatment surveillance. It is important that PtDAs and personalized risk information are embedded in the clinical workflow and effectively used to support information provision and decision-making [11]. The literature shows that people struggle to understand risk information, irrespective of educational level [12,13,14,15]. For effective risk communication, risk communication guidelines need to be applied [16,17].

The effectiveness of the BCS-PtDA was evaluated in the SHOUT-BC study [18,19]. The results of the SHOUT-BC study showed that the use of the BCS-PtDA led to a higher level of patient-reported SDM and that patients perceived their role in the decision-making process as more active. Furthermore, patients found the BCS-PtDA and the use of risk information regarding personalized risks for LRRs and SPs relevant and useful [19].

To ensure long-term (i.e., out-of-study) implementation and up-scaling, potential facilitators and barriers relating to the BCS-PtDA, the users, and the organization need to be identified [11,20]. This can help identify potential improvements to BCS-PtDA application and its impact on risk communication and the SDM process in clinical practice.

Therefore, we carried out a process evaluation focusing on (1) the observed implementation and participation rates, (2) facilitators and barriers for HCPs’ use of the BCS-PtDA, (3) the observed level of SDM, and (4) how risk information was communicated and how SDM was applied in doctor–patient consultations.

## 2. Methods

### 2.1. Study Design

Mixed methods were used to perform the process evaluation. This evaluation was embedded within the SHOUT-BC study (see Figure 1 for a graphic overview of the SHOUT-BC study) [10,18,19]. This study was conducted in accordance with local laws and regulations. The Medical research Ethics Committees United in Nieuwegein, the Netherlands, has confirmed that the Medical Research Involving Human Subjects Act (WMO) does not apply to this study (reference number W19.154).

### 2.2. Breast Cancer Surveillance Decision Aid

The BCS-PtDA was developed in cocreation with relevant stakeholders and consists of three components [10]: (1) a handout sheet with which post-treatment surveillance can be introduced and on which personalized risks for recurrences (estimated using the INFLUENCE 2.0 nomogram, https://www.evidencio.com/models/show/2238/nl (accessed on 30 December 2023)) can be visualized, (2) a web-based deliberation tool including patient information and the 6-item Cancer Worry Scale [21], and (3) a summary sheet that can support the final decision-making during the clinical consultation (Figure 2).

### 2.3. Measures and Procedures

Table 1 provides an overview of the measures and methods used for the four aims of this study.

#### 2.3.1. Aim 1: Assessing the BCS-PtDA Implementation and Participation Rate

The implementation rates were calculated by dividing the total number of patients that received the BCS-PtDA by the total number of patients meeting SHOUT-BC inclusion criteria (Figure 1) diagnosed at participating hospitals between November 2019 and December 2021 (i.e., post-implementation period). The participation rate was calculated by the number of patients who logged into the BCS-PtDA divided by the number of patients who were issued the BCS-PtDA.

The total number of patients eligible to receive the BCS-PtDA was estimated using data of participating hospitals from the Netherlands Cancer Registry (NCR). The total number of patients that received the BCS-PtDA and the number of patients that logged into the BCS-PtDA was based on log data retrieved from ZorgKeuzeLab (the organization that is involved in the development, maintenance, and implementation of the BCS-PtDA). We also retrieved log data regarding which parts of the BCS-PtDA patients visited, the number of visits, and average duration of visits.

#### 2.3.2. Aim 2: Identifying Facilitators and Barriers of PtDA Use by HCPs

All HCPs (*n* = 47) involved in the implementation of the BCS-PtDA were sent a survey about the implementation of SDM supported by risk information approximately six months after the start of the implementation phase at their hospital. We used a selection of the Measurement Instrument for Determinants of Innovations (MIDI) questionnaire to identify facilitators and barriers related to the innovation (7 determinants), user (35 determinants), and organization (7 determinants) [20]. Responses were scored using Likert scales ranging from 1 (e.g., “totally disagree”) to 5 (e.g., “totally agree”). In line with a previous study, MIDI determinants that were answered by ≥20% of HCPs with “totally disagree/disagree” (or “false” or “not a single colleague, almost no colleague, a minority”) were considered barriers, and items answered by ≥80% with “agree/totally agree” (or “true” or “a majority, almost all colleagues, all colleagues”) were considered facilitators [22]. With the questionnaire, we also collected data about sociodemographic characteristics of the HCPs.

#### 2.3.3. Aim 3: Quantifying the Observed Level of SDM

To quantify the level of SDM after implementation of the BCS-PtDA, consultation(s) of women who participated in the SHOUT-BC study in which the organization of post-treatment surveillance was discussed were audio-recorded with the patients’ consent. The audio-recorded consultations were transcribed verbatim. The level of SDM during consultations was determined using the 5-item OPTION scale [23]. OPTION-5 is a validated scale that evaluates components of SDM, namely (1) the presentation of options, (2) establishment of patient partnership, (3) pros and cons of the options, (4) elicitation of patient preferences, and (5) integration of patient preferences into the decision. Each item is rated from 0 (absence of SDM competency) to 4 (optimal performance). As four separate decisions are presented in the BCS-PtDA, we scored the OPTION-5 for each decision separately. Two researchers (FLR and EE) coded the OPTION-5 for all consultations. Discrepancies were resolved through consensus. The OPTION-5 score was calculated by summing up the score on each item, then dividing the resulting sum score by the maximum achievable score (i.e., 20), and, finally, multiplied by 100. The score reflects the percentage of the maximum achievable score that was achieved in each consultation. The higher the OPTION-5 score, the greater the level of observed SDM. The median scores for each decision modality considered in the consultations were calculated.

#### 2.3.4. Aim 4: Qualitatively Assessing Risk Communication and SDM Application in Doctor–Patient Consultations after Implementation of the BCS-PtDA

To analyze the quality of the content of the audio-recordings, we used a combined technique of deductive and inductive thematic analysis. Our initial coding framework for the thematic content analysis was based on the 4-step SDM model [7]. We started with a predefined coding framework of the main themes. Emergent subthemes were identified from the data during coding and added to our framework. To assess the quality of the communication of personalized risk estimates from the INFLUENCE 2.0 nomogram, we incorporated components of the BRISK scale [17] into the coding framework. The BRISK scale is a brief observational measure of clinical risk communication competence consisting of four items: (1) description of the reference class (patient group), (2) use of contrasting frames to describe the risk estimates (e.g., framing as risk of recurrence vs. disease-free survival), (3) use of absolute terms to describe the risk differences (including use of percentages vs. natural frequencies to present information), and (4) acknowledgement of the uncertainty (i.e., aleatory and epistemic uncertainty) associated with the risk estimates. In our analysis, we focused on the qualitative execution of each competency (i.e., what is done well and what needs to be addressed to foster effective communication) and not on quantification of competence. Finally, we also noted the duration of consultations.

All transcripts were double coded by two researchers (FLR and EE). Discrepancies were resolved through consensus. Qualitative data analyses were performed using MAXQDA2007 software. To be able to put the findings emerging from the content coding into perspective, two informal sessions were organized with clinicians participating in the SHOUT-BC study to discuss the preliminary results. The findings were used to contextualize our observations.

## 3. Results

### 3.1. Aim 1: Assessing the BCS-PtDA Implementation and Participation Rate

In the eight participating hospitals, 26% of the total of 1834 eligible women received the BCS-PtDA, of whom 61% logged in to the BCS-PtDA. The implementation and participation rates strongly varied over the hospitals (Table 2).

Patients spent a median of 26 min reviewing the BCS-PtDA (range: 1 to 156 min) and visited the BCS-PtDA a median of 1.7 times (range: 1 to 7 times). Each of the information pages for the various topics were accessed by between 77% and 92% of the patients. Further, 89% of participating patients completed the statements to help them clarify their values in the BCS-PtDA (i.e., values clarification exercises), between 84 and 87% indicated their preference for all four post-treatment surveillance options discussed in the BCS-PtDA, and 85% indicated their preferred role in decision-making in the BCS-PtDA. Finally, 86% completed the 6-item Cancer Worry Scale.

### 3.2. Aim 2: Identifying Facilitators and Barriers of BCS-PtDA Use by HCPs

The HCP’s survey response rate was 24 out of 47 invited (63%). It was completed by at least two professionals from each participating hospital. Two-thirds of the respondents were female surgical oncologists with an average age of 47 years (SD = 9) and, on average, 11 years (SD = 7) of experience with breast cancer care. The handout sheet was provided by ten participating HCPs to up to five patients, eight provided it to between five and ten patients, and six provided it to ten to twenty-five patients.

Table 3 provides an overview of HCPs’ responses to MIDI determinants representing potential facilitators and barriers to the implementation of the BCS-PtDA (see Appendix A for a complete overview). Facilitators related to the intervention were HCPs’ perception that (1) the BCS-PtDA is based on factually correct knowledge, (2) all information and materials needed to work with the BCS-PtDA are provided, and (3) that the BCS-PtDA is not too complex to use. A barrier related to the intervention was the perceived lack of observable outcomes of BCS-PtDA use (e.g., a decrease in intensity of surveillance appointments).

Related to the potential users of the BCS-PtDA (i.e., patients and HCPs), facilitators were (1) HCPs indicated the BCS-PtDA helps to create choice awareness and (2) that they could count on adequate assistance from colleagues to use the BCS-PtDA. Barriers were HCPs’ perceptions that their use of the BCS-PtDA (1) would not reduce the time needed to inform patients, (2) would not provide more time to discuss patients’ considerations and preferences, (3) would not decrease their workload, and (4) would not be adequately supported by their superior.

Related to the organizational aspects, one facilitator identified “having a coordinator to manage the process of implementing the BCS-PtDA” and four barriers: (1) not having enough time to integrate the BCS-PtDA into day-to-day work, (2) other changes going on, (3) lack of regular feedback about progress, and (4) lack of formal ratification by management.

### 3.3. Aim 3: Quantifying the Observed Level of SDM

In total, the OPTION-5 was scored for the consultations of 64 participants (Appendix A provides the participant characteristics). The median total duration of the sum of both consultations in which the BCS-PtDA and the post-treatment surveillance decision was discussed was 19 min (range: 7–88 min). There was wide variation in the duration of the consultations both within and between hospitals. The OPTION-5 scores showed a suboptimal level of SDM across all four decisions regarding the organization of post-treatment surveillance (Table 4).

### 3.4. Aim 4: Qualitatively Assessing Risk Communication and SDM Application in Doctor–Patient Consultations after Implementation of the BCS-PtDA

Qualitative content analysis was performed on the same 64 post-implementation consultations for which the OPTION-5 was scored. The consultations mainly focused on post-treatment surveillance. Little time was spent on aftercare-related topics. Cancer worry was a frequently discussed topic in the consultations, most often triggered by the discussion of the mammography findings. Below, we discuss the results per each step of SDM. See Table 5 for example quotes.

#### 3.4.1. Creating Choice Awareness

In most consultations, no clear reason was stated why patients have a choice regarding the organization of post-treatment surveillance. Often, only after discussing the purpose of the SHOUT-BC study did this become clear. HCPs generally framed the rationale for the evaluation study and offering women a choice as a way to reduce the number of visits for post-treatment surveillance and the associated anxiety/cancer worry. In almost all consultations, HCPs accentuated that the patient’s preferences would guide decision-making (see Table 5 relating to SDM step 1 section A for an example quote). Furthermore, HCPs indicated that patients should feel free to choose whichever option best matched their preferences and circumstances.

Unintended implicit steering behavior was observed. Some HCPs told patients what they/the medical team generally would advise them to do or what the guideline recommends. Decision-making was framed as we would normally advice you to do X, but you have a say in this and could choose to do something else (see Table 5 relating to SDM step 1 section B for an example quote).

#### 3.4.2. Information Provision

##### Presentation of Options

The BCS-PtDA provided patients with information about the four choices (frequency, maximum duration, type of tests performed, and how the post-treatment surveillance results would be communicated). Generally, not all choices were discussed. The most frequently mentioned options were (1) the frequency of mammographic screening and physical examination and (2) whether the patient would come to the hospital to discuss the results. There usually was a lack of discussion about potentially relevant considerations (e.g., pros and cons) for the different options (see Table 5 relating to SDM step 2 section A for example quotes). Patients’ comprehension was rarely checked, besides occasional general questions such as “Is it clear?”, “Do you have any questions?”, or a statement such as “Please interrupt if anything is unclear or if you have questions”.

#### 3.4.3. Risk Communication

None of the participating patients were asked whether they wanted to receive prognostic estimates from the INFLUENCE 2.0 nomogram in the audiotaped consultations. The nomogram was usually briefly introduced. In general, the predictors were not explained. HCPs adhered to some of the basic risk communication rules (e.g., using natural frequencies and mentioning the time horizon). However, the reference class was vaguely described or not at all, and risks were mostly communicated using a negative frame (i.e., chance of disease recurrence without mentioning chance that the disease would not recur). Also, at times, jargon was used to communicate risk information, which made it difficult to comprehend (i.e., ipsilateral recurrence or contralateral breast cancer or a new primary tumor; see Table 5 relating to SDM step 2 section B for an example quote). Patients and, at times, HCPs alluded to the lifetime risk of developing breast cancer for women in the general population and attempted to make a comparison to the recurrence risk.

Further, general allusions were regularly made to the uncertainty associated with the probabilities, mainly related to aleatory uncertainty (i.e., the inability to predict future events; see Table 5 relating to SDM step 2 section C for an example quote). Although the INFLUENCE 2.0 nomogram provides information about epistemic uncertainty (expressed as a confidence interval around the point estimate), this was infrequently discussed. And, if mentioned, it was usually in the form of a general statement referencing that “it is statistics”.

Finally, during the discussion about the outcome estimates with HCPs, patients mainly asked clarifying questions to check whether they had understood the risk information correctly. Many questions seemed to be an attempt to qualify the probabilities—is this a high percentage, and should I be worried?

#### 3.4.4. Deliberation and Decision-Making

Overall, there was a lack of deliberation during the final decision-making. HCPs usually only mentioned patients’ choices listed on the BCS-PtDA summary sheet as a check. Generally, there was no discussion about what patients’ considerations were (i.e., why they preferred a specific option) (see Table 5 relating to SDM step 3 and 4 section A for an example quote). In informal sessions with HCPs in which the preliminary results were discussed, we probed why they did not deliberate with patients. Attendees indicated that they did not want to give patients the impression that they had to justify themselves or that they had made a wrong choice. However, when patients indicated that they wanted to receive the results of the tests via telephone, HCPs explicitly checked whether patients understood that it meant they would forego the yearly physical breast examination (see Table 5 relating to SDM step 3 and 4 section B for an example quote). In some instances, HCPs steered patients away from a telephone consultation. In the informal sessions, some HCPs indicated to be hesitant to forego physical examinations (e.g., because some patients might not be proficient enough in breast self-examination). Other HCPs felt the physical examination had little added value and only probed patients’ preferences to make sure they understood.

We observed that, often, no explicit decision was made for all four decisions presented in the BCS-PtDA. HCPs did often clearly indicate that the patient could always revisit their choice. In the informal session with HCPs to discuss the preliminary results, HCPs voiced disappointment in the lack of a reduction in surveillance consultations due to the use of the BCS-PtDA, which made them question the added value of the BCS-PtDA.

## 4. Discussion

In this process evaluation, we observed a low overall implementation rate (19–42%) and modest to good participation rates (29–83%), with variations between hospitals. Quantitative analysis of the SDM levels using OPTION-5 showed consistently poor performance on SDM across all four decisions regarding the organization of post-treatment surveillance (median score range: 15–50 points out of 100). The qualitative content analysis showed that deliberation about post-treatment surveillance options was scarce. Risk communication was deemed generally adequate. HCPs observed that the BCS-PtDA effectively increased patient awareness about their treatment options. However, they reported concerns about an increased workload and that they were uncertain about the added value of the PtDA.

HCPs did not use the BCS-PtDA with all eligible patients (19–42% implementation rate). When provided to patients, it was used by, on average, 61% of patients, and most patients viewed all of the BCS-PtDA components. These findings potentially suggest that patients value the BCS-PtDA more highly than HCPs. Not many studies on PtDAs report implementation rates, and if such rates are reported, varying metrics are used, making comparisons difficult. Cuypers et al. [24] evaluated the implementation rates following the introduction of three different PtDAs for prostate cancer treatment across multiple regions in the Netherlands (in 33 hospitals). They found an overall implementation rate of 40%, with a range from sporadic usage (less than 10%) to high implementation (over 80%) for all three PtDAs. Although our implementation rates fall within the observed range, the average rate of 26% can be considered low. Foremost, these low implementation rates stem from barriers indicated by the HCPs’ questionnaire results, such as a lack of time available for integration of the BCS-PtDA into day-to-day work or a perceived lack of support from supervisors and management. Our findings also underscore the importance of involving all relevant stakeholders (patients, health care professionals, management, etc.) early on in the development and implementation process to achieve broad support for implementation [11,25], including senior-level participation to ensure resources are in place to facilitate implementation [11].

The MIDI results reflected a mismatch between the aim of SDM (i.e., better informed patients making value-congruent choices) and HCPs’ expectations about the outcome of using the BCS-PtDA (e.g., reduction in the number of surveillance appointments). Disappointment in a lack of a tangible effect on the intensity of surveillance appointments might significantly impact HCPs’ motivation for (long-term) use of the BCS-PtDA. Although we observed a change in intensity of the frequency of surveillance appointments in the SHOUT-BC study, it was only a slight decrease [19]. Therefore, it is important to make sure that HCPs have realistic expectations about the goal of the BCS-PtDA. Further, the lack of objective deliberation, as observed in the consultations, also may lead to HCPs not having a clear understanding of patients’ rationale for their choice and whether revisiting the decision should be put on the agenda for future surveillance appointments.

The suboptimal levels of SDM that were observed using OPTION-5 contradict the levels of SDM experienced by patients in the SHOUT-BC effectiveness study, which showed significant increases in SDM after introduction of the BCS-PtDA [19]. This illustrates the importance of not only assessing end-users’ perceptions but combining these with observations like in this process evaluation. The low OPTION-5 scores observed in our process evaluation are in line with the literature [8]. The clear mismatch between the observed and perceived levels of SDM seen in the SHOUT-BC data has also been reported in other studies (e.g., [26,27]). An explanation for this finding is that patients take different factors into account (e.g., satisfaction with the clinician) [28], meaning a mismatch in the construct being measured by perceived and observed SDM instruments (“self-report bias”). It is important to create awareness of these discrepancies among HCPs using PtDAs in different settings.

The qualitative analysis of the content of the consultations identified SDM areas requiring improvement. Specifically, not all the options in the BCS-PtDA were explicitly discussed with patients. Also, there was a lack of elaboration on potentially relevant considerations when the options were discussed. A possible explanation for the succinct presentation of the options is that HCPs viewed the BCS-PtDA as a replacement for information provision during the consultation. And therefore, they chose to only briefly give examples of the options patients were informed about. However, PtDAs are not intended to replace information provision by HCPs. Another area requiring improvement was the presence of unintended steering. This can unintentionally undermine clinicians’ intent of involving patients in the decision-making process. Unintended steering is of particular concern, as we also observed that there was a lack of deliberation. Deliberation is a crucial step in SDM; thus, extra attention should be given in future SDM training on how HCPs can effectively and objectively deliberate with their patients, considering them as partners rather than subjects. To further support HCPs in applying deliberation, cues (like example sentences) can be incorporated into the BCS-PtDA.

In the assessment of the quality of the content, we also looked at risk communication in the consultations. We observed that personalized risks were communicated to patients without ascertaining whether they wished to receive this information. To avoid forcing risk information on patients, it is important to perform a check. Although HCPs adhered to the basic risk communication guidelines [16,29], the wording used to describe the outcome being presented was not always clear. The literature shows that people, irrespective of educational level, struggle to understand risk information [12,13,14,15]. Therefore, clarity of presentation is key to ensure patients understand this information correctly and needs to be given extra attention in risk communication training for HCPs. Since there was little interaction regarding the risk information, misunderstandings are likely to have gone unnoticed.

### 4.1. Strengths and Limitations

Our study is one of few studies focusing on evaluation of the process of implementation of a PtDA, including a content analysis of consultations. A key strength is the assessment of PtDA implementation and SDM with risk information using a large sample of audio recordings from clinical consultations. This provided valuable insights into actual usage, impact on communication, and the areas requiring improvement. Our study also has some limitations. Firstly, for practical reasons, it is impossible to know exactly how many patients were given access to the BCS-PtDA. The total number of patients who received the BCS-PtDA is an estimate based on the login identification number of the women who did use the BCS-PtDA. As the login identification number on the consultation sheet blocks are continuous, the assumption is made that all numbers between login identification numbers that were used to log in were also handed out to patients. This could be an overestimation, meaning that the actual implementation rate might be lower and the actual participation rate might be higher. Secondly, the reported outcomes from the HCP’s survey might not be generalizable to all HCPs, as those who completed the survey might have been more supportive of the implementation of the BCS-PtDA (“self-selection bias”). However, the variation in scores suggests the survey data did capture variations in the views and preferences among HCPs. Thirdly, obtaining audio recordings for the study was difficult (with one hospital not recording any audios) (“selection bias”), and incomplete recordings made during the post-implementation phase could have led to us potentially missing parts of the SDM process. Finally, we did not include questions on negative financial consequences of implementing the BCS-PtDA in the survey. A reduction in surveillance diagnostics and appointments (i.e., reduction in production) means a reduction in income for HCPs. Therefore, this might also be a barrier to implementation.

### 4.2. Practice Implications

Our findings highlight the need for continuous evaluation after PtDA implementation to consistently improve and facilitate long-term effectiveness. Although HCPs underwent training on SDM application, risk communication, and the utilization of the BCS-PtDA, our findings indicate the need for additional training to ensure effective SDM implementation and adequate risk communication. Further, successful SDM implementation requires a fundamental change in behavior from both HCPs (doctors, nurse practitioners, and management alike) and patients. Full integration into routine health care processes is essential (e.g., allotment of dedicated SDM time and reimbursement). It is therefore vital to raise awareness among HCPs about the positive outcomes resulting from the initial time investment in adopting SDM, such as well-structured consultations (i.e., more effective and efficient use of consultation time) and better informed patients. Finally, implementation efforts should empower and remind patients to engage in SDM and to make use of available decision support.

## 5. Conclusions

Overall, the implementation rates of the BCS-PtDA were low. The participation rates were modest to good. When the BCS-PtDA was used during consultations, patients were clearly given a choice regarding their post-treatment surveillance plan. However, there are areas for improvement in terms of use of the BCS-PtDA, risk communication, and application of SDM to foster effective patient engagement in decision-making about post-treatment surveillance. This study underscores that process evaluations are needed, alongside effectiveness studies reporting patient evaluations, to broaden the knowledge regarding the successful long-term implementation of PtDAs and effective SDM.

## Figures and Tables

**Figure 1 cancers-16-01390-f001:**
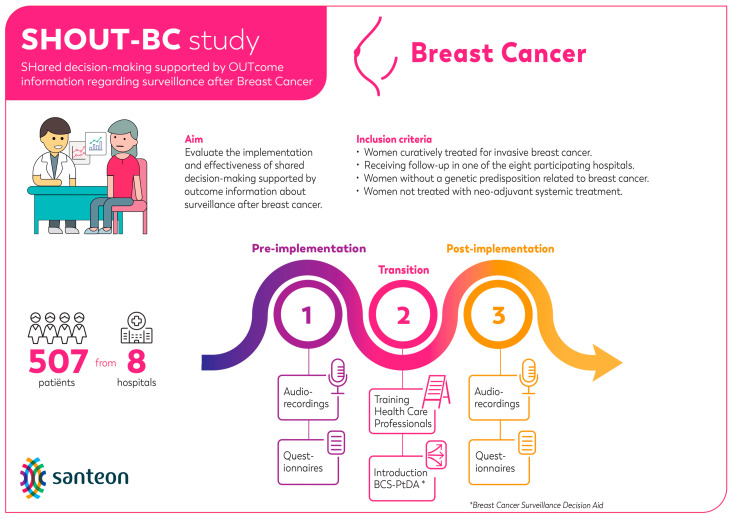
Graphic overview of the SHOUT-BC study.

**Figure 2 cancers-16-01390-f002:**
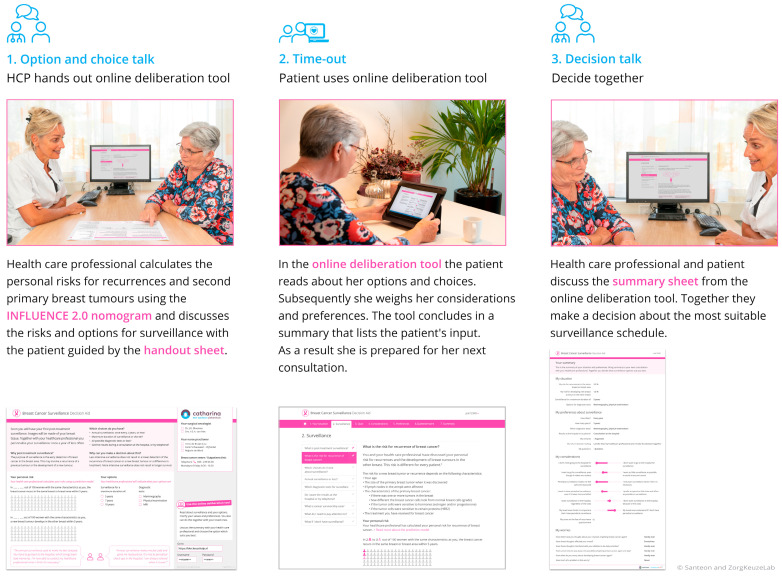
The components of the Breast Cancer Surveillance Decision Aid.

**Table 1 cancers-16-01390-t001:** Overview of the methods and measures used.

Research Aims	Measures	Methods	Sample
**Aim 1 Assessing the BCS-PtDA implementation and participation * rate.**	**Implementation rate:** 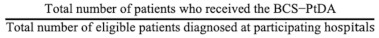 **Participation rate:** 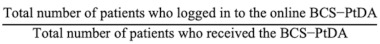 The implementation and participation rates were calculated overall and per hospital.	**Descriptive analysis of registry and log data:** -Total number of patients eligible to receive the BCS-PtDA was estimated using data extracted from the NCR. The SHOUT-BC inclusion criteria were applied in the extraction procedure.-Total number of patients that received the BCS-PtDA and the total number of patients that logged in to the BCS-PtDA was based on log data retrieved from ZorgKeuzeLab.	Registry data from women meeting the SHOUT-BC inclusion criteria who were treated at a participating hospital and log data from women who were issued the online BCS-PtDA.
**Aim 2 Identifying facilitators and barriers of BCS-PtDA use by HCPs.**	**MIDI:** The innovation, user, and organization scales of the MIDI questionnaire were used to obtain insights into the determinants affecting the implementation of the BCS-PtDA (original version, in Dutch) [20]. MIDI items that were answered by ≥20% of HCPs with “totally disagree/disagree” were considered barriers, and items answered by ≥80% with “agree/totally agree” were considered facilitators [22].	**Descriptive analysis of survey data:**Data collected using an online survey.	Survey data from all HCPs who could hand out the BCS-PtDA in hospitals participating in the SHOUT-BC study.
**Aim 3 Quantifying the observed level of SDM after implementation of the BCS-PtDA.**	**Observed level of SDM:** The OPTION-5 scale [23] was used.	**Quantitative analysis of consultation transcripts:**Consultations were double-coded using the OPTION-5 scale. Differences in coding were resolved through consensus.	Transcripts of consultations with women participating in the SHOUT-BC study.
**Aim 4 Qualitatively assessing risk communication and SDM application in doctor–patient consultations after implementation of the BCS-PtDA.**	**Qualitative content analysis consultations:** a self-developed coding framework based on the 4-step SDM model by Stiggelbout et al. [7] and the BRISK scale components [17] was used. We also coded the duration of the consultation.	**Qualitative analysis of consultation transcripts:**Consultations were double coded using a combined technique of deductive and inductive thematic analysis. Differences in coding were resolved through consensus.	Transcripts of consultations with women participating in the SHOUT-BC study.

BCS-PtDA: Breast Cancer Surveillance Decision Aid; NCR: Netherlands Cancer Registry; SHOUT-BC: SHared decision-making supported by OUTcome information regarding surveillance after curative treatment for Breast Cancer; HCP: Health Care Professional. * As part of the evaluation of the participation rate, we also assess the extent of use of the BCS-PtDA (i.e., duration of visits and which pages patients visited).

**Table 2 cancers-16-01390-t002:** Implementation and participation rates of the BCS-PtDA per hospital.

Hospital	Number of Patients Eligible to Receive BCS-PtDA (A)	Number of Patients Who Received the BCS-PtDA (B)	Number of Patients Who Logged in to the BCS-PtDA (C)	Implementation Rate (B/A, %)	Participation Rate (C/B, %)
1	245	103	73	42	71
2	265	51	38	19	75
3	306	65	54	21	83
4	318	67	22	21	33
5	245	95	49	39	52
6	373	70	48	19	69
7	82	27	12	33	44
8	N/A	14	4	N/A	29
**Total**	**1834**	**492**	**300**	**26**	**61**

BCS-PtDA: Breast Cancer Surveillance Decision Aid.

**Table 3 cancers-16-01390-t003:** MIDI determinants for use of the BCS-PtDA by HCPs (*n* = 24).

Determinants Associated with the Innovation (BCS-PtDA)			
	Totally Agree/ Agree (%)	Neutral (%)	Totally Disagree/ Disagree (%)
Procedural clarity: it is clear which activities I should perform when offering the BCS-PtDA	79	8	13
Correctness: the BCS-PtDA is based on factually correct knowledge	** 88 **	4	8
Completeness: all information and materials to work with the BCS-PtDA properly are provided	** 83 **	4	13
Complexity: the BCS-PtDA is **not** too complex for me to use ^a^	** 83 **	13	4
Compatibility: the BCS-PtDA is a good match for how I am used to working	58	25	17
Observability: the outcomes of using the BCS-PtDA are clearly observable	29	33	** 38 **
Relevance for patient: the BCS-PtDA is relevant for my patients	58	33	8
**Determinants Associated with the Users (i.e., Patients and HCPs)**			
	Totally agree/ Agree (%)	Neutral (%)	Totally disagree/ Disagree (%)
Personal benefits: the BCS-PtDA saves me time in informing my patients	0	29	** 71 **
Personal benefits: the BCS-PtDA provides me more time to discuss the considerations and preferences of my patients with them	25	25	** 51 **
Personal drawback: my workload has not increased by using the BCS-PtDA ^a^	8	25	** 67 **
Outcome expectations (importance): the BCS-PtDA helps to create awareness that there is a choice regarding the organization of post-treatment surveillance	** 92 **	4	4
Outcome expectations (importance): the BCS-PtDA helps to inform about and discuss the different options regarding the organization of post-treatment surveillance	79	13	8
Outcome expectations (importance): the BCS-PtDA helps to clarify the wishes and preferences of my patients regarding the organization of post-treatment surveillance	75	17	8
Outcome expectations (importance): the BCS-PtDA helps to make a shared decision regarding the organization of post-treatment surveillance	79	17	4
Professional obligation: I feel that it is my responsibility to use the BCS-PtDA	63	21	17
Patient satisfaction: patients are generally satisfied when I use the BCS-PtDA	54	29	17
Patient cooperation: patients generally cooperate when I use the BCS-PtDA	58	33	8
Social support: I can count on adequate assistance from my colleagues if I need it to use the BCS-PtDA	** 83 **	17	0
Social support: I can count on adequate assistance from my superior if I need it to use the BCS-PtDA	54	46	** 25 **
	A majority, Almost All Colleagues, All Colleagues (%)	Half of Colleagues (%)	Not a Single Colleague, almost No Colleague, a Minority (%)
Descriptive norm: proportion of colleagues that actually use the BCS-PtDA ^b^	67	17	17
**Determinants Associated with the Organization (i.e., hospital)**			
	Totally Agree/ Agree (%)	Neutral (%)	Totally Disagree/ Disagree (%)
Time available: there is enough time available to integrate the BCS-PtDA as intended in my day-to-day work	25	17	** 58 **
Material resources and facilities: there are enough materials and facilities provided to use the BCS-PtDA as intended	71	17	13
Information accessible: it is easy for me to find information about using the BCS-PtDA as intended	75	17	8
Performance feedback: feedback is regularly provided about progress with the implementation of the BCS-PtDA	50	21	** 29 **
	**True**		**False**
Formal ratification by management: there are formal arrangements relating the use of the BCS-PtDA ^c^	33		** 33 **
Coordinator: one or more people have been designated to coordinate the process of implementing the BCS-PtDA ^c^	** 88 **		4
Unsettled organization: there aren’t any other changes going on that could influence implementation of the BCS-PtDA ^a,c^	25		** 58 **

Abbreviations: HCP: health care professional, BCS-PtDA: Breast Cancer Surveillance Decision Aid, and PROM: patient-reported outcome measure. MIDI determinants that were answered by ≥20% of HCPs with “totally disagree/disagree” were considered barriers, and items answered by ≥80% with “agree/totally agree” were considered facilitators. Numbers underscored on the left represent a HCP reported a facilitator, and numbers underscored on the right represent a HCP-reported barrier. ^a^ Determinant is reversed for readability/interpretability. ^b^ Answer categories were divided into (1) “a majority, almost all colleagues, all colleagues”; (2) “half of colleagues”; and (3) “not a single colleague, almost no colleague, a minority”. ^c^ Possible answer categories were divided into “true” (=facilitator) and “false” (=barrier); answering option “I don’t know” was not considered as part of the categorization as a facilitator or barrier.

**Table 4 cancers-16-01390-t004:** OPTION-5 scores per decision modality (*n* = 64).

	Median (Range)
Frequency of screening	25 (20–45)
Duration of post-treatment surveillance	15 (0–30)
Type of screening examinations performed	25 (0–45)
In-person consultation vs. receiving results over the telephone	25 (0–50)

Notes. OPTION-5 is a validated scale that evaluates the components of SDM: (1) presentation of options, (2) establishment of patient partnership, (3) pros and cons of options, (4) elicitation of patient preferences, and (5) integration of patient preferences into the decision. Each item is rated from 0 (absence of SDM competency) to 4 (optimal performance). OPTION total scores are calculated by summing the five items. The sum score is then divided by the maximum achievable score (5 × 4 = 20) and multiplied by 100. The score indicates the percentage of the maximum score that was achieved. The maximum score achievable is 100, and higher scores reflect a better application of SDM during the consultation. The median score and range are calculated per decision.

**Table 5 cancers-16-01390-t005:** Example quotes from the consultations for each step of SDM.

SDM STEP 1—CREATING CHOICE AWARENESS		
Section A Accentuating that there is a choice and that it should be guided by patients’ preferences		“[When it comes to follow-up checks] you can make choices and we don’t want to steer you. It’s about you being aware of the risk [prognostic implications] and making a choice that suits you”. “Well, normally, by default, we do it [post-treatment surveillance] the same way for everyone. But not every person is the same and not every person has the same risk. That’s the reason we try to study if it makes sense to educate people [about their personalized risk]… so that they can then make their own personal consideration whether to adjust the follow-up, depending on their [personalized] risk, given their wishes…”.
Section B Possible (implicit) steering due to framing of choice and referencing personal advice or guidelines		“Normally we [clinicians] always say about the first two years, come back every six months… Unless you would like otherwise”. “Based on this outcome [risk estimates from the INFLUENCE 2.0 nomogram], I recommend five years of monitoring. That means five years here at the hospital, with a mammogram and a physical examination annually. We check for a total of 10 years, only then [the latter five years] we do it through the family physician”. “Well, how do we deal with that [post-treatment surveillance] as doctors before the study [SHOUT-BC study]? We obviously have guidelines in the Netherlands that we adhere to. Guidelines are advice, advice to doctors. Well, the advice with breast cancer like you also had is, that you remain under surveillance for five years. That’s actually the advice. But I also have had people before this study opened, with whom I discussed this [personalizing post-treatment surveillance], not with these fancy numbers [risk estimates from INFLUENCE 2.0 nomogram]. … to whom I’ve also said, is there a point to the follow-up? And a number of people have said, well, thank you very much for operating on my breast, but I’ll see you when I feel something again. Of course, those are primarily elderly patients”.
**SDM STEP 2—INFORMATION PROVISION**		
Presentation of options	Section A Succinct presentation of options	Well, based on that [risk estimates from the INFLUENCE 2.0 nomogram] you can think, how often do I really have to actually come here [to the hospital]? Especially if that causes a lot of stress. Every year? Once every two years or maybe even less often? And the maximum period [of the post-treatment surveillance], in principle, five years. Or a shorter period? Do all the tests have to be done or can you do with a little less? The results you can get in the hospital or by phone. Those are the choices.
Communication of prognostic estimates	Section B Presentations of risk information—unclear which outcome is being discussed	You can see the graphic here. It shows 100 women with the same characteristics [as you]. Then you see that within five years after surgery, two out of 100, so 98 don’t get it [breast cancer] back. Two get the disease back. So the chances of not getting it [breast cancer] back are much higher. Then if you look at a hundred women, same age, et cetera. And look who of those got breast cancer again within five years, that’s three out of a hundred women, so 97 don’t get it back. … And then here you see the risks per year. So that’s around a one percent per year chance. And this [risk per year] always adds up.
	Section C Discussion of aleatory uncertainty (i.e., inability to predict future events)	Well, in order to indicate a little bit what it could mean for you, we can also use statistics. We are fond of that [using statistics], aren’t we, in the medical world, to express everything in numbers. And, of course, it’s not like I have a crystal ball and can predict what will happen to you, but we can make an estimate, can’t we?
**SDM STEP 3 and 4—DELIBERATION AND DECISION-MAKING**		
Section A Discussion about current choice and potential for change of preference over time		Patient: Look, maybe if I know this longer [a longer period of time] that it’s [okay to have fewer check-up] then I might say oh well, maybe once every two years or so [is fine]. … HCP: Totally fine, totally fine. Patient: But maybe once I’ve processed it, that I’ll think like oh, well [less check-ups is okay] HCP: Exactly. And if maybe after two check-ups, you think well, it’s okay [no disease recurrence] for a second time, I feel a lot calmer about it all, then that [reconsidering frequency of check-ups] can be discussed. But so this [discussion] is really to make you aware [of the risk and the options].
Section B Deliberation about whether to opt for a telephone consultation to discuss the results		Patient: Yes. I’m not anxious, I’m very down-to-earth, but thinking about [whether to get a mammogram], then I want to know [whether the disease has recurred], if I can. HCP: Totally understandable, indeed. Now you indicated [on the BCS-PtDA summary sheet] that you want to get the results during a telephone consultation. Patient: Yes, that was fine by me, you know. HCP: But does that means that you don’t want a physical examination? So you keep track of that yourself [do breast self exams], or do you say, I want a physical examination as well? Because that part [of the post-treatment surveillance], if we annually discuss the results over the phone, means that the physical examination is dropped. So that might be something to think about, what you want. And then you say, I would like the health care provider to choose. That’s not what the decision aid is for, of course, precisely to ensure that you make your own choices. I can advise you, though. Patient: That’s what I mean. HCP: And again, the advice, I also put that at the top here, would be to at least check for five years with mammography and a physical examination. You may also do that physical examination at the doctor’s office. If you say, I monitor that very well myself and I’ll get in touch if I feel anything, that’s fine too. Patient: I never actually do that [breast self exams], I must admit. I don’t know what to feel for. I happened to feel this [the breast tumor] myself, but that was because it was just so noticeable… HCP: We can also do it together to see how skillful you can become in doing a breast exam. That I tell you what I’m doing and that I show you what to look for yourself. And if you say, I feel confident enough with that, then you do it yourself, and if that’s not the case, you just come here. Yes? Patient: Yes. HCP: Nice. I’m going to make a note of the choice you entered [on the BCS-PtDA summary sheet]. You can deviate from it [your choices] at any time. If at some point you say, but I’d like an extra check or I’d like to skip a check, that’s always good. It’s about doing what makes you happy. As long as we communicate about that with each other, that’s the most important thing.

Note. BCS-PtDA: Breast Cancer Surveillance Decision Aid; HCP: health care professional; SDM: Shared decision-making; SHOUT-BC study: SHared decision-making supported by OUTcome information regarding surveillance after curative treatment for Breast Cancer study.

## Data Availability

The data that support the findings of this study are available from the corresponding author, J.W., upon reasonable request.

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
