# Peer review of "Evaluation of the Implementation of the Dutch Breast Cancer Surveillance Decision Aid including Personalized Risk Estimates in the SHOUT-BC Study: A Mixed Methods Approach"

_cancers, 2024, doi:10.3390/cancers16071390_

Round 1

Reviewer 1 Report

Comments and Suggestions for Authors

Jet W. Ankersmid et al. evaluate the implementation of the Dutch Breast Cancer Surveillance Decision Aid, including personalized risk estimates, in the SHOUT-BC study using a mixed methods approach. The hypothesis and sim developed by the authors are in sync with the conclusions in this work.

The authors should explain how they were able to avoid bias in data collection.

Most of the literatures used in this article is not recent. The literature review should be extended and improved.

Comments on the Quality of English Language

Minor editing of English language required

Author Response

Comment 2.1: Jet W. Ankersmid et al. evaluate the implementation of the Dutch Breast Cancer Surveillance Decision Aid, including personalized risk estimates, in the SHOUT-BC study using a mixed methods approach. The hypothesis and sim developed by the authors are in sync with the conclusions in this work.
Authors’ response to comment 2.1: Thank you for your time and effort and feedback on our manuscript.

Comment 2.2: The authors should explain how they were able to avoid bias in data collection.
Authors’ response to comment 2.2: We were not able to completely avoid bias in data collection. In the limitations paragraph in the discussion section (lines 426 to 433) we have described the most important types that we faced and their potential impact on our findings. We have made the types of biases more explicit in the text to guide readers to find this information. We have also described the mismatch between perceived and observed SDM in the discussion section. This mismatch could be due to ‘self-report bias’ and we’ve now explicitly mentioned that in the text (line 384).

Comment 2.3: Most of the literatures used in this article is not recent. The literature review should be extended and improved.
Authors’ response to comment 2.3: Thank you for your comment. We recognize that some of the references are not so recent. The literature that we’ve cited are the seminal works in this field. To our knowledge, this is the state-of-the-art. We have reviewed our literature and have updated where necessary.

Comment 2.4: Minor editing of English language required

Authors’ response to comment 2.4: Thank you for alerting us. We have reviewed the manuscript and have made minor textual edits.

Reviewer 2 Report

Comments and Suggestions for Authors

Dear authors,
Many thanks for submitting your work to the journal. Your manuscript is well-written and well-structured. My review revealed the following few minor issues:

• Please clarify if all of the questionnaires included in your study have been previously translated and validated in the Dutch language.

• Please provide a comprehensive synopsis of the main findings of your study in the first paragraph of the discussion for readers' convenience.

• Define abbreviations in every table of your study.

Best regards

Author Response

Comment 3.1: Many thanks for submitting your work to the journal. Your manuscript is well-written and well-structured.
Authors’ response to comment 3.1: Thank you for your comments. We appreciate your time and effort in providing us with feedback on this manuscript.

Comment 3.2: Please clarify if all of the questionnaires included in your study have been previously translated and validated in the Dutch language.
Authors’ response to comment 3.2: Thank you for raising this point. The MIDI questionnaire that was used for this study was developed in the Netherlands and was therefore originally available in Dutch. We have made this more explicit in the manuscript in Table 1.

Comment 3.3: Please provide a comprehensive synopsis of the main findings of your study in the first paragraph of the discussion for readers' convenience.
Authors’ response to comment 3.3: We have rewritten the first paragraph of the discussion section to provide a more comprehensive overview of the main findings (lines 336 to 345).

Comment 3.4: Define abbreviations in every table of your study.
Authors’ response to comment 3.4: Thank for addressing this. We have made sure that all abbreviations are defined in the tables.
